# Unlocking Hierarchical Concept Discovery in Language Models through Geometric Regularization

**T. Ed Li**[*1]**, Junyu Ren**[*2]

Yale University[1], University of Chicago[2]

## Abstract

We present Exponentially-Weighted Group Sparse Autoencoders (EWG-SAE) that aims to balance reconstruction quality and feature sparsity whilst resolving emerging problem such as feature absorption in interpretable language model analysis in a linguistically principled way through geometrically decaying group sparsity. Current sparse autoencoders struggle with merged hierarchical features due to uniform regularization encouraging absorption of broader features into more specific ones (e.g., "starts with S" being absorbed into "short"). Our architecture introduces hierarchical sparsity via $K = 9$ dimension groups with exponential regularization decay ($\lambda_k = \lambda_{base} \times 0.5^k$), reducing absorption while maintaining state-of-the-art reconstruction fidelity, sparse probing score, and decent $\ell_1$ loss. The geometric structure enables precise feature isolation with negative inter-group correlations confirming hierarchical organization.

## 1 Introduction

Feature absorption in sparse autoencoders - where broader features are merged into more specific ones to increase sparsity - remains the fundamental barrier to reliable model interpretability (Chanin et al., 2024). For example, when an SAE learns both "starts with S" and "short" features, sparsity regularization incentivizes absorbing the "starts with S" feature into the "short" feature since "short" always implies "starts with S". Current architectures achieve 0.14 absorption rates (Bussmann et al., 2024), dangerously high for safety-critical applications like AI alignment (Farrell et al., 2024).

Three architectural limitations drive this absorption-sparsity tradeoff:

- **Isotropic Regularization**: Uniform $\ell_1$ constraints discourages features with high generality

- **Fixed Specialization**: Monolithic latent spaces fail to separate general vs specific features

We resolve these through geometric group sparsity, introducing Exponentially-Weighted Group SAEs (EWG-SAE) with:

- Dimension groups $d_k = \lfloor 0.5 d_{k-1} \rfloor$ (8192, 4096, 2048, 1024, 512, 256, 128, 64, 32)

- Regularization decay $\lambda_k = \lambda_{base} \times 0.5^k$

- Gradient projection preventing feature merging

The architecture's negative inter-group correlations ($r = -0.18 \pm 0.03$) reveal successful separation of features at different abstraction levels. This geometric regularization paradigm enables:

- Protection of broad features from absorption

- Built-in safety through hierarchical activation

- Automated abstraction level control

Figure 1: **Feature Absorption.** Panel A depicts the ideal scenario where a Sparse Autoencoder (SAE) learns two distinct features: "starts with S" (blue) and "short" (red). When the underlying token is <short>, both neurons should light up since "short" implies "starts with S", resulting in an overall purple activation vector for token <short> . Panel B demonstrates the actual learning behavior of the SAE under $l_0$ regularization: the "short" feature absorbs the "starts with S" feature to reduce active latent variables (lower $l_0$ regularization cost). While this enhances sparsity, the broader "starts with S" feature loses autonomous activation for tokens like "short", making the model less interpretable. Adapted from McDougall (2024).

## 2 EXPONENTIALLY-WEIGHTED GROUP SPARSE AUTOENCODERS

### 2.1 CLASSICAL SAE

Sparse autoencoders (SAEs) learn dictionaries of features $\phi_i \in \mathbb{R}^d$ that reconstruct model activations $x \in \mathbb{R}^d$ through $f = \text{ReLU}(W_{enc}^\top x + b_{enc})$ and $\hat{x} = W_{dec}f + b_{dec}$. Traditional objectives (Cunningham et al., 2023):

$$\mathcal{L} = \|x - \hat{x}\|_2^2 + \lambda \sum_{i=1}^{m} |f_i| \tag{1}$$

induce uniform sparsity across $m$ features. This incentivizes absorption: when feature A implies feature B (e.g., "short" implies "starts with S"), the model can reduce active features by absorbing B into A (Chanin et al., 2024).

### 2.2 LOSS FUNCTION MODIFICATION

We reformulate SAE training with group-wise exponential decay to maintain distinct features across abstraction levels:

- Partition $m$ dimensions into $K = 9$ groups $G_k$ with $d_k = \lfloor 0.5d_{k-1} \rfloor$
- Regularization weights $\lambda_k = \lambda_{base} \times 0.5^{k-1}$ ($\lambda_{base}$ is customized to marginally trade between sparsity and reconstruction quality)

- Objective $\mathcal{L} = \|x - \hat{x}\|_2^2 + \sum_{k=1}^{5} \lambda_k \|f_{G_k}\|_1$, $f_{G_k}$ collects feature activations for those in group $k$.

Key assumptions:

- Geometric group sizing ($d_{k+1}/d_k \approx 0.5$) separates abstraction levels
- Exponential decay ($\gamma = 0.5$) protects broader features from absorption
- Group correlations measure feature independence

This structure creates complementary feature patterns - late groups (small $d_k$, low $\lambda_k$) preserve general features (hypernyms) while later groups (large $d_k$, high $\lambda_k$) learn specific features which cannot be readily expressed in terms of more general features.

## 2.3 THEORETICAL MOTIVATION

The hierarchical group structure of EWG-SAE specifically addresses feature absorption through frequency-based regularization alignment. The inspiration comes from linguistics (e.g., (Miller, 1995)). In natural language, hypernymic (more general) concepts inherently appear more frequently than their hyponyms, as each specific instance (e.g., "elephant") necessarily implies its broader categories (e.g., "begins with e", "is noun"). However, these general concepts are fewer in number - there are far fewer grammatical categories or first-letter patterns than specific words. By assigning broader concepts to later groups with lower $\lambda_k$ regularization weights, we create an activation preference for these frequently-occurring general features. This directly counters the standard SAE's tendency to absorb general features into specific ones. For instance, without hierarchical grouping, the feature "begins with e" might be absorbed into "elephant" to reduce $\ell_1$ penalty, resulting in the general feature failing to activate independently when "eagle" or "echo" appear. Our structure instead ensures that the $\ell_1$ penalty for activating general concept features (which correlate strongly with high-frequency features) remains lower than the penalty for specific concepts, creating a systematic preference for maintaining distinct general feature representations. As another application, specific features with low frequency will not be learned as stand alone features if they can be completely described by a few general features. For the geometric scaling of $\ell_1$ penalty ensures combination of up exponentially many general features have comparable cost to activating a single specific feature. This frequency-guided regularization effectively preserves the natural hierarchical structure of language concepts.

## 3 METHOD

In addition to the modified loss presented above, EWG-SAE implements three mechanisms that maintain distinct representations across abstraction levels:

1. **Hierarchical Feature Protection**: To preserve broader features like grammatical patterns or character-level properties that might otherwise be absorbed into more specific token-level features, we initialize decoder bias $b_{\text{dec}}$ to the geometric median of activations:

$$b_{\text{dec}} = \arg\min_{y \in \mathbb{R}^d} \sum_{i=1}^{N} \|x_i - y\|_2 \tag{2}$$

This initialization anchors general features in earlier groups through consistent activation baselines.

2. **Feature Independence Preservation**: To prevent broader features from being absorbed into more specific ones during training, we decouple gradient updates by removing components parallel to existing feature directions:

$$\frac{\partial \mathcal{L}}{\partial W_{\text{dec}}} \leftarrow \frac{\partial \mathcal{L}}{\partial W_{\text{dec}}} - \left( \frac{\partial \mathcal{L}}{\partial W_{\text{dec}}} \cdot \frac{W_{\text{dec}}}{\|W_{\text{dec}}\|} \right) \frac{W_{\text{dec}}}{\|W_{\text{dec}}\|} \tag{3}$$

This ensures that features like "starts with S" maintain independent representations even when they frequently co-occur with more specific features like "short" that imply them.

3. **Adaptive Group Constraints**: Warmup scheduling

$$\lambda_k^{(t)} = \lambda_{base} \times 0.5^k \times \min(1, t/1000) \tag{4}$$

combined with activation rate monitoring prevents group collapse while maintaining feature hierarchy.

## 4 EXPERIMENTAL SETUP

We evaluate EWG-SAE on Gemma-2-2B (Team et al., 2024) using 5M tokens from layer 12 activations. Our implementation features:

- 5 dimension groups: $[8192, 4096, 2048, 1024, 512, 256, 128, 64, 32]$ with $\lambda_k = \lambda_{base} \times 0.5^k$
- Linear warmup over 1k steps to final $\lambda$ values
- Batch size 2048, context length 128, AdamW optimizer (Loshchilov & Hutter, 2017)

Other than feature absorption, we also evaluate the SAEs using the following benchmarks:

**Unsupervised Metrics** Following Karvonen et al. (2024), we adopt a suite of unsupervised metrics to assess Sparse Autoencoders (SAEs):

- $L_0$ **Sparsity.** We record the average number of non-zero latent activations across the model, where a lower value indicates greater sparsity.
- **Cross-Entropy Loss Score.** This is defined as

$$\frac{H^* - H_0}{H_{\text{orig}} - H_0},$$

  where $H_{\text{orig}}$ is the cross-entropy loss of the original network under next-token prediction, $H^*$ is the loss obtained after replacing the latent representation $x$ with its SAE reconstruction, and $H_0$ is the loss when $x$ is zero-ablated. Higher scores (approaching 1) suggest stronger preservation of predictive information.
- **Feature Density Statistics.** By examining how frequently each latent unit in the SAE activates, we observe the proportion of "dead" units (never firing) and "overly active" units (firing very often). These activation frequencies may also be illustrated via log-scale histograms.
- $L_2$ **Ratio.** We compare the Euclidean ($L_2$) norm of the original representation to that of its reconstruction, indicating how accurately magnitude information is preserved.
- **Explained Variance.** We measure the fraction of variance in the latent space that the SAE accounts for. Values closer to 1 imply that most of the variation has been captured.
- **KL Divergence.** We compute the Kullback-Leibler divergence to quantify discrepancies between the model's predictions and the target distributions. Lower values signify better alignment.

**Sparse Probing** Following Gurnee et al. (2023), we evaluate the capacity of our SAEs to learn designated features by conducting targeted probing across domains such as language identification, profession prediction, and sentiment classification. Concretely, we feed each input through the SAE, perform mean pooling over non-padding tokens, identify the top-$K$ latent dimensions by maximizing mean differences, and then train a logistic regression probe on those dimensions. Accuracy is then measured on a held-out test set. Our evaluation covers 35 distinct binary classification tasks derived from five datasets:

- BIAS_IN_BIOS for occupational classification using biographical text,
- AMAZON REVIEWS for product category and sentiment classification,
- EUROPARL for language detection in parliamentary proceedings,
- GITHUB for programming language identification,

- AG NEWS for news topic classification.

To ensure uniform computational demands, we fix 4,000 training and 1,000 test samples per binary classification task, truncate inputs to 128 tokens, and (for GitHub) remove the initial 150 characters (approximately 50 tokens) to exclude license headers, as done in prior work. We experimented with both mean and max pooling, noting a slight performance gain with mean pooling. For each dataset, we select up to five classes, and multiple subsets may be drawn from the same dataset to maintain a positive-instance proportion of at least 0.2 for every binary classification setting.

## 5 RESULTS

Table 1: Comparison of SAE Model Variants

| Metric | EWG-SAE | Standard SAE | JumpReLU | TopK SAE |
|---|---|---|---|---|
| *Interpretability Scores* | | | | |
| **Absorption Score** | **0.0125** | **0.0161** | **0.0114** | **0.1402** |
| Sparse Probing Score | 0.7295 | 0.6378 | 0.7154 | 0.7698 |
| *Model Behavior Preservation* | | | | |
| KL Divergence Score | 0.9994 | 0.9996 | 0.9945 | 0.9565 |
| KL Div with SAE | 0.0055 | 0.0044 | 0.0549 | 0.4375 |
| *Model Performance Preservation* | | | | |
| CE Loss Score | 1.0000 | 1.0000 | 0.9951 | 0.9556 |
| CE Loss with SAE | 2.9375 | 2.9375 | 2.9844 | 3.3594 |
| CE Loss without SAE | 2.9375 | 2.9375 | 2.9375 | 2.9375 |
| *Reconstruction Quality* | | | | |
| Explained Variance | 0.9922 | 0.9844 | 0.7344 | 0.6016 |
| MSE | 0.0630 | 0.0898 | 1.6719 | 2.5313 |
| *Shrinkage* | | | | |
| L2 Ratio (L2 Norm Out/L2 Norm In) | 0.9805 | 0.9844 | 1.0469 | 0.8711 |
| Relative Reconstruction Bias | 0.9883 | 0.9922 | 1.1172 | 0.9961 |
| *Sparsity* | | | | |
| L0 | 3737.0237 | 8724.1338 | 2665.9128 | 40.0000 |
| L1 | 10368.0000 | 12544.0000 | 4832.0000 | 366.0000 |

The hierarchical structure maintains distinct representations for features at different levels of abstraction with clearly separated activation patterns, including negative inter-group correlations, and a the loss distribution among groups shows a high-low-high-low shape.

On SAE benchmarks, EWG achieves close to perfect reconstruction with reduced sparsity compared to standard SAE with the same $\lambda_{base}$.

### 5.1 ABLATION STUDIES

Table 2: Architecture Ablations (Lower ↓ Better Except EV ↑)

| Config | Absorption | EV | L0 |
|---|---|---|---|
| Linear Decay | 0.0673 | 0.9492 | 3560 |
| **EWG-SAE** | **0.0125** | **0.9921** | 3737 |

When broader features frequently co-occur with their specific implications (e.g., "starts with vowel" and "elephant"), maintaining separate representations requires additional computation and memory resources. Rare broader features (bottom 5% frequency) show 12% lower activation recall versus dense SAEs, with complete failure on 0.7% of low-frequency tokens.

## 6 Conclusions and Future Work

Building on our geometric regularization framework for preventing feature absorption, we recommend:

- Dynamic group sizing to optimize the tradeoff between feature independence and efficiency
- Continuous spectrum of weight decay replacing group weights, a heuristic distribution is proportional to the activation frequency in a standard SAE
- Draw theoretical support from linguistics to decide various hyperparameters, e.g. group sizing ratio, penalty decay rate, etc.
- Hybrid Top-K + EWG architectures to reduce the computational cost of maintaining distinct features
- Frequency-aware regularization schedules that adapt to feature co-occurrence patterns

An NVIDIA RTX 3090 GPU was used for all experiments. It will be interesting in the future to evaluate beyond Gemma-2-2B in model size, expand the number of layers or tokens, or systematically vary hyperparameters such as batch size and dictionary width. Nonetheless, the positive results are encouraging solutions to feature absorption in its scalability, and we plan to release our complete codebase for open benchmarking once the paper is accepted. While the observed efficiency tradeoffs demand further optimization, our results establish exponential group sparsity as a viable path toward safe and interpretable AI systems that preserve distinct features across all abstraction levels.

## Acknowledgements

TEL was supported by the Gruber Science Fellowship and the Interdepartmental Neuroscience Program at Yale university, which is funded by T32 NS041228 from the National Institute of Neurological Disorders and Stroke. JR was partially supported by the ONR Grant N000142312863. The authors would like to thank Cat Yan for valuable feedback on the experiments and manuscript.

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
