# OpenReview forum: "UNLOCKING HIERARCHICAL CONCEPT DISCOVERY IN LANGUAGE MODELS THROUGH GEOMETRIC REGULARIZATION"
_ICLR.cc/2025/Workshop/BuildingTrust — BuildingTrust_

### Official Review · Reviewer_1iys · 2025-02-19
**Good idea but too similar to Matryoshka SAEs; Reject**

**Rating:** 4
**Confidence:** 4

**Review:**

This paper introduces EWG-SAEs, which are constructed to learn hierarchical features by introducing various levels of SAE size and sparsity penalties. This approach is intended to aid with feature absorption, where independent concepts are combined into a single feature to “hack” the sparsity penalty of SAEs.

The main idea of this paper seems to be identical to Matryoshka SAEs, published on LessWrong on December 13th [1] and 19th [2]. This paper does not acknowledge Matryoshka SAEs, and does not consider them as a parallel work to compare EWG-SAEs to.

In general, I think this is a strong idea with good theoretical underpinnings and is well evaluated. However, there is no acknowledgement of Matryoshka SAEs and no comparison of methods, resulting in a lack of novelty. Thus, I vote to reject.

Strengths
- Uses nice motivations from linguistics to motivate the use of heirarchial SAEs as an analogy to heirarchial concepts in natural language
- Constructs an intuitive SAE architecture to take advantage of this insight
- The feature absorption is clearly better with an EWG-SAE than other methods
- I like the use of feature independence preservation as a decoupled gradient update in Equation 3, it seemed effective

Weaknesses
- Biggest weakness is the lack of novelty due to the similarity to Matryoshka SAEs. There should be a discussion of how EWG-SAEs measure up against Matryoshka SAEs and a comparison of their theoretical underpinnings. From my understanding, these methods are virtually identical.
- Table 1 contains almost all of the key results of the paper, yet is very difficult to read. Please bold the best value in each row so it is easy to assess how EWG-SAE's stackup from a glance. It is currently far too many unmarked numbers to be digestible.
- While the paper claims to use ideas from linguistics, the main idea seems to be that concepts are heiarchial. It would be good to justify the use of hyperparemeters, like the 0.5 exponential decay rate in width using linguistics ideas.
- There is a lack of hyperparameter evaluation. How does performance change as the number of groups d increases? How sensitive are results to the exponent used in the sparsity function or the width decay rate?

Minor points
- I think section 2.3 should come before section 2.2 (motivation before analytics)

Questions
- I'm confused by the Model Performance/Behavior preservation sections of Table 1. For example, what is CE Loss Score, CE Loss Score with SAE, and without SAE? I don't think it is described in the paper.
- In 2.2 says that d_k = 0.5 d_k-1. But later says that d_k+1/d_k = 0.75. Why the discrepancy?


[1] https://www.alignmentforum.org/posts/zbebxYCqsryPALh8C/matryoshka-sparse-autoencoders
[2] https://www.lesswrong.com/posts/rKM9b6B2LqwSB5ToN/learning-multi-level-features-with-matryoshka-saes

---

### Official Review · Reviewer_cYeX · 2025-02-22
**Simple and effective technique at improving SAE interpretability with grouped latents and geometrically decaying regularization**

**Rating:** 7
**Confidence:** 3

**Review:**

This paper introduces a novel training methodology for Sparse Autoencoders (SAEs) designed to improve interpretability by reducing feature absorption. The approach draws inspiration from linguistics, specifically the observation that broader concepts tend to appear more frequently than specific ones. The authors propose a structured organization of the SAE's latent space, dividing the latent dimensions into a series of "groups" of decreasing size. Smaller groups are intended to capture broader, more frequently activated concepts, while larger groups represent more specific, less frequently activated features.

Another key contribution is a modified regularization scheme. The authors apply a geometrically increasing regularization penalty across the groups. Smaller, "broader" groups receive a smaller penalty, encouraging their activation, while larger, "more specific" groups receive a stronger penalty.

The authors evaluate their approach using a series of benchmarks. They report achieving state-of-the-art feature absorption scores, suggesting improved interpretability, and demonstrate that the general model performance remains comparable to standard SAE training. These results will certainly be useful for the mechanistic interpretability community but it would be preferred if more information is included and presentation is improved.

Strengths:
- Modifies the training of SAEs that enables low feature absorption scores leading to better interpretability while maintaining performance
- The modification is also quite simple and can easily be used in practice without needing specialized libraries.
- Good justification from the linguistics literature for the usage of exponential decay of activations in groups

Weaknesses:
- Unclear why 5 groups were used for experiments. It would be beneficial to test for other values of K as well
- How was the correlation calculated? The regular pearson correlation is defined for two variables but instead a single correlation is given for all 5 groups.
- Unclear how activation rate is measured. In particular, how exactly are the broad and specific features defined? Some examples are provided in the text but this is presumably not the whole list.
- A table is provided for the ablation study but there is no description in the text that mentions what exactly is being tested. It seems like it is measuring the difference between using linear decay and exponential decay but we do not know how the linear decay factor for the groups.
- (minor) Paper mentions 3 limitations which lead to the sparsity absorption tradeoff but only 2 are provided (line 34).
- (minor) The Gemma model is cited as "Team et al" but should preferably be "Gemma Team"
- (minor) I'm assuming that the sigma in the results (lines 255, 257) refers to the standard deviation but it might be better to be more explicit
- (minor) Possible typo on line 113. It is mentioned that d_{k+1} / d_k = 0.75 but this should be 0.5 if the same dimensions are used as in line 42 where each group is half the size of the preceding group.

Comments:
- I was initially very confused with the usage of the word "groups" and thought that the paper would be using group theory from mathematics. It might be beneficial to use a different term to describe the separation of latents in the SAE. Perhaps a figure would also be useful in explaining the architecture.

---

### Decision · Program_Chairs · 2025-03-05

**Decision:**

Accept

**Comment:**

This paper introduces Exponentially-Weighted Group Sparse Autoencoders (EWG-SAEs) to improve feature sparsity and reduce feature absorption in SAEs. The approach applies hierarchical sparsity with exponentially increasing penalties, inspired by linguistic principles. While the idea is interesting and well-evaluated, Reviewer 2 (R2) points out that Matryoshka SAEs, a closely related prior work, is not cited or compared against. Comparing with this prior-art would be helpful.